# Community characteristics and the risk of non-communicable diseases in Ghana

**Winfred A. Avogo** [ORCID] *

Department of Sociology and Anthropology, Illinois State University, Normal, Illinois, United States of America

* wavogo@ilstu.edu

## Abstract

Non-communicable Diseases (NCDs) are rising quickly in low- and middle- income countries. In Ghana, chronic diseases are major causes of morbidity and mortality, yet data and the evidence- base for awareness, detection, and management of NCDs are lacking. Using data from the 2014 Ghana Demographic and Health Survey (GDHS), the first national study with information on hypertension and other risk factors, we examine the correlates and community characteristics associated with the risk of hypertension, obesity, and anemia among women. We find that hypertension prevalence in Ghana was 16 percent and 17 percent were overweight/obese, while 41 percent had anemia of any form. On community characteristics, the level of poverty in a community was significantly associated with lower risks of all three NCDs, while the aggregate level of employment had higher risks. On individual characteristics, the wealth of a household, women's educational level and urban residence were significant predictors of NCDs. We interpret the findings within the literature on neighborhood characteristics, the social gradient of health and in the context of speeding up the attainment of the Sustainable Development Goals (SGDS) to reduce premature deaths by one-third by 2030.

## Introduction

According to research by World Health Organization (WHO), non-communicable diseases (NCDs), which includes cardiovascular diseases, diabetes, cancer, chronic respiratory illnesses, and mental health disorders, are the leading cause of death worldwide, collectively responsible for 74 percent of global mortality [1]. An estimated 41 million people die each year from NCDs. Nearly 75 percent of all NCD deaths occur in low-and middle-income countries. Similarly, 86 percent of people who die from NCDs before reaching their seventieth birthday, die in low-and middle-income countries [1]. In sub-Saharan Africa (SSA), deaths from NCDs are rising faster than anywhere else in the world and we now know that the on-going COVID-19 pandemic has the highest morbidity and mortality risks among older adults especially those with underlying conditions such as NCDs [2].

In Ghana, NCDs account for 43 percent of all deaths, with cardiovascular diseases accounting for 19 percent of NCD deaths [3]. The few population-based surveys available on Ghana have also shown that NCDs are growing among the urban poor with a dual burden of

**Data Availability Statement:** Data can be obtained at https://dhsprogram.com/Data/.

**Funding:** The authors received no specific funding for this work.

**Competing interests:** The authors have declared that no competing interests exist.

infectious and chronic diseases. The prevalence of hypertension (raised blood pressure), for example, has been increasing over several decades and has significant impact on cardiovascular disease morbidity and mortality, especially in urban Ghana [4–8]. Similarly, there is a high and rising prevalence of obesity among Ghanaian adults–over 43 percent of the adult population are either overweight or obese [9]. Also, anemia which disproportionately affects children, women, and individuals from low-income areas of the country is associated with many chronic conditions, including Human Immunodeficiency Virus (HIV) and sickle cell disease as well as malaria and other cognitive and physical performance conditions [10]. All three chronic risk factors of NCDs (hypertension, obesity, and anemia) are asymptomatic in nature (silent killers), especially during the early stages when interventions and treatments are most effective.

The economic and social impact of NCDs is attributable to rapid demographic, epidemiological and nutritional transitions that are occurring in SSA and pose a threat to the attainment of the 2030 Agenda for Sustainable Development (SGDs), which includes a target (3.4) to reduce by one third, [relative to 2015 levels] premature mortality from NCDs and to promote mental health and well-being [11]. Currently, more than half of countries in the world are likely to miss SDG target 3.4 [12]. NCDs also threaten the attainment of the demographic dividend- the eventual aging of the population which results in fewer dependent children and the elderly and therefore, greater economic productivity through labor force participation. Yet, despite these threats, routine data collection systems and the broad evidence- base for increased awareness, detection, management, and control of NCDs are lacking in Ghana and other resource constrained settings.

In this paper, we draw on the first nationally representative sample of women aged 15–49 from the Demographic and Health Surveys (DHS) that include high-quality data on NCDs to examine the influence of neighborhood/community-level characteristics and conditions and the risks of NCDs with special focus on prevalence of hypertension, overweight/obesity, and anemia in Ghana. We interpret the results within the literature on neighborhood characteristics and health and in the context of interventions to accelerate the attainment of SDG target 3.4. to improve health, reduce death and disability due to NCDs and to improve the prospects of a demographic dividend in sub-Saharan Africa.

The epidemiological, demographic, and sociological literature has seen an explosion in research on the effect of neighborhoods and the community context on health [13–15]. This is, in part, a recognition that social influences on health operate through several structural conditions that shape individual lives and opportunities [16].

Studies on the relationship between neighborhood characteristics and NCD risk factors in higher income countries suggest a possible influence of community characteristics on health [17]. For example, living in communities with higher socio-economic disadvantage is associated with higher Body mass index (BMI), net of individual characteristics [18]. Similarly, individuals living in zip codes with the greatest percentage of happy and physically active tweets had lower obesity and diabetes prevalence net of individual characteristics [19]. Indeed, these findings are supported by the theory of the social gradient of health which posits that people of lower socio-economic status (SES) suffer a heightened health risk for nearly all diseases (including NCDs) compared to those of higher SES [20–23]. However, the social gradient of health hypothesis may be less consistent in low-to middle-income countries at the initial-middle stages of the demographic and epidemiological transitions as lifestyle and behavior changes of the emerging middle class may predispose them to diseases including NCDs. Thus, some studies in developing countries have shown a negative association (higher SES increase the risk of NCDs) [24–26] while others have demonstrated a positive association (higher SES decrease the risk of NCDs) [27,28]. Given these emerging socio-economic changes and disparities in individual risk factors of NCDs, as well as the dearth of broad evidence on the impact

of neighborhoods on NCDs in low-middle-income countries, there is the need to explore the mechanisms that link neighborhood characteristics to NCD risk in these settings.

We draw on two mechanisms to conceptualize this relationship. First, diffusion theory highlights the role of social learning and social influence through individual social interaction in the community [29–33]. Social learning and social influence through education and employment levels in a community may afford women the information and awareness of NCDs, lifestyle changes and appropriate health seeking behaviors to avoid the risk of NCDs. Social influence can offer social and emotional support that buffers the risk of stress and NCDs [34]. Likewise, healthy behaviors such as access to and consumption of healthy food, health screening, not smoking, moderate alcohol consumption etc. are known to spread through social learning and social influence within social networks [35]. These spillover effects from other women's education and awareness of NCDs and behaviors may affect the behavior of uneducated women in communities with high aggregate levels of education compared to uneducated women elsewhere [36].

On the other hand, neighborhood effects could impact NCD risks negatively through the globalization and diffusion of unhealthy lifestyles. The broader social and economic transformation which has led to a rise in average community education and employment may have sped up the ongoing nutritional and epidemiological transition thereby leading to poor dietary habits from processed Western diets and foods high in calories, sugar, salt, and total fat [37] as well as the lack of physical exercise and immoderate alcohol consumption. These habits, which are peculiar to the affluent, educated, middle class in urban centers are known to increase the prevalence of obesity, hypertension, and diabetes. Thus, women living in communities with high levels of aggregate education, wealth and employment will be impacted by the risk of NCDs independent of their individual education and employment.

Finally, Ghana's health care system despite the decentralization of health services to the community-level through the Community-based Health Planning and Services (CHPS) is still relatively weak- basic healthcare services for the detection and screening of NCDs are inaccessible nor are those services targeted towards the poor [38]. Thus aggregate-level health coverage in a community in terms of the National Health Insurance Scheme (NHIS) and health seeking behavior in the community will impact NCD risks at the population level. Poor women who reside in communities with high aggregate-level health coverage will have low risks factor burdens of NCDs.

## Materials and methods

### Data

Data are drawn from the cross-sectional study of the Ghana Demographic and Health Survey (GDHS) conducted in 2014, the first survey in Ghana to include several anthropometric and blood pressure measurements, Body Mass Index (BMI) and anemia testing. The analysis relies on a total of 9396 women aged 15–49. Women who did not have information on their blood pressure readings, as well as pregnant women were excluded from the measurement of hypertension, obesity, and anemia. The 2014 GDHS was conducted by the Ghana Statistical Service (GSS), the Ghana Health Service (GHS), and the National Public Health Reference Laboratory (NPHRL) of the GHS in collaboration with the Measure DHS+ program of the United States Aid for International Development (USAID). The survey was designed to assist program managers, researchers and policymakers involved in planning, managing, and coordinating strategies for improving the health of Ghanaians.

The MEASURE DHS+ program also has experience with addressing ethical issues related to the protection of human subjects during biological sampling. Ethical clearance was provided

by the Ghana Health Service Ethical Review Committee, Research and Development Division, Ghana Health Service; and the Institutional Review Board of ICF International. Descriptions of the protocols for biological testing can be found in a separate documentation on the DHS website (e.g., Anemia Testing Manual for Population-Based Surveys [39].

The DHS surveys used a two-stage sample design to produce estimates of key indicators at the national level. In the first stage, stratified sampling techniques are used to select clusters, delineated from 2010 Ghana Population and Housing Census (PHC), as the primary sampling unit (PSU). This resulted in a total of 427 clusters selected in the entire country (216 clusters in urban areas and 211 clusters in rural areas).

The second stage involved a systematic random sampling of households within each PSU. About 30 households were selected from each cluster and a total of 12,831 households were selected throughout the country. Since the sample is not self-weighting at the national and sub-sample levels, weighting factors (provided in the data) are used to produce results that are proportional at the national level and account for unequal probability of selection. On the response rate, 97 percent of the eligible women in each household were interviewed [40].

## Variable measurement

**Dependent outcomes.** *A. Hypertension.* Blood pressure measurements were taken from consenting women aged 15–49 at intervals of 10 minutes during the individual interview. Blood pressure was measured using the LIFE SOURCE UA-767 Plus blood pressure monitor: a digital oscillometric blood pressure measuring device with automatic upper-arm inflation and automatic pressure release. In this paper, hypertension was defined as an average of the second and third measurements of systolic blood pressure (SBP) $>=$ 140 mmHg and/or an average diastolic blood pressure (DBP) $>=$ 90 mmHg according to internationally recommended categories [41]. Blood pressure measurements excluded pregnant women at the time of the survey due to the possibility of gestational hypertension.

*B. Obesity.* The GDHS survey also includes height (in cm) and weight (in kg) measurements for women aged 15–49. Weight and height measurements were used to estimate body mass index (BMI = weight in kg $\div$ [height in meters, $^2$]). BMI was further classified as underweight (BMI $<$ 18.5 kgm$^{-2}$), normal weight (BMI = 18.5–24.9 kgm$^2$), overweight (BMI = 25.0–29.9 kgm$^{-2}$), and obese (BMI $\geq$ 30.0 kgm$^{-2}$) according to WHO recommendations. Pregnant women at the time of the survey were excluded since their BMI, comparable to other women in the survey, should be based on their weight before pregnancy.

*C. Anaemia testing.* Blood specimens for anemia testing were collected in half of the selected households from women aged 15–49 who voluntarily consented to be tested. Blood samples were drawn from a drop of blood taken from a finger prick and the haemoglobin concentration was measured using the HemoCue photometer system. Hemoglobin levels were further categorized into severe anemia if less than 7.0 g/dl and moderate between 7 and 9.9 g/dl and mild 10 and 12.9 g/dl and not anemic (13 g/dl and above), according to WHO recommendations [42]. Anemia was recoded as a dichotomous outcome (1- any form of anemia vs 0- not anemic). Pregnant women were also excluded from the measurement of anemia.

## Independent variables

**Community variables.** We use three variables to capture neighborhood/aggregate levels of education, employment, and poverty in the community. The first is the percentage of educated women in the community. This variable was constructed by aggregating the individual and household level variables at the primary sampling unit (PSU). The second aggregate level variable is the percentage of working women in the community. Finally, we include the

percentage of poor women in the community. These three community variables have been demonstrated to be associated with reproductive outcomes [36,43].

**Covariates.** The DHS surveys collected information on the characteristics of respondents at the time of the survey. We used individual educational attainment (1. No education, 2. Primary education, Secondary or higher education) as a covariate to examine the impact of community variables on the dependent outcomes. Individual women's education has been shown in numerous studies to have a strong effect on health outcomes [44]. We included variables on the place of residence (urban vs. rural) to capture the effect of urbanization, the age-group of respondents is also included as the risk of NCDs varies by age [37,38]. The region of residence (Southern vs. Northern Ghana) is accounted for to capture inequalities in socio-economic development across the country, ethnicity (Akan, which is the largest ethnic group, vs other tribes) is included as a covariate to account for differences in morbidity and access to healthcare that are tied to ethnic background. The wealth index of the household, which is constructed using Principal Component Analysis, is included as a composite measure of household wealth. It combines information on household water supply and sanitation, floor type and electricity as well as other household goods into an index [45]. The index is recoded from quintiles into two categories of 1. Poorest and poorer respondents and 2. Middle, richer and richest respondents. We also include National Health Insurance coverage (no, yes) as a proxy for healthcare access and visits to health facilities in the last 6 months (yes, no).

## Analytical methods

Descriptive statistics were used to produce cross tabulations and chi-square tests of independence to describe variations in the dependent outcomes. Multilevel binary logistic regression was used to examine the impact of community variables and individual covariates on hypertension, obesity, and anemia. Due to the hierarchical nature of the data, (individual women are nested within clusters and households), failure to control for the correlation resulting from characteristics of women within the same cluster and household as well as aggregating the community variables at the cluster level will result in biased standard errors and estimates. We applied appropriate sampling weights (level weights) that correspond to each stage of sampling in DHS surveys at both the bivariate level and multivariate level. Results show that omitting level-weights may lead to underestimating the variation between level-2 units (at the cluster level and household level).

At the multivariate level, we fitted multilevel logistic regression models using XTMELOGIT command in STATA. Two-level random intercept models from the two-stage sampling design with the PSU and sampling weights were applied. We also fit Survey Logistic procedure in STATA: "svy: logit" to compare the results. Both procedures produced identical results. For more information on multilevel modelling using DHS data see [46]. A check for multicollinearity was conducted using the variance inflation factor (VIF) and all the variable used had less than 10 (VIF) and tolerance value lower than 1.0 (1/VIF) suggesting the absence of multicollinearity.

## Ethical considerations

Procedures and questionnaires for standard DHS surveys have been reviewed and approved by ICF Institutional Review Board (IRB). Additionally, country-specific DHS survey protocols are reviewed by the ICF IRB and typically by an IRB in the host country. ICF IRB ensures that the survey complies with the U.S. Department of Health and Human Services regulations for the protection of human subjects (45 CFR 46), while the host country IRB ensures that the survey complies with laws and norms of the nation. Written informed consent was obtained from the parent/guardian of each participant under 18 years of age.

# Results

## Descriptive results

Table 1 shows the percent distribution of NCDs, community variables and background characteristics of respondents. The table shows 16 percent of women were considered hypertensive (16.6 percent) and a similar percentage were considered overweight/obese (17 percent) while 41 percent reported anemia of any form.

**Table 1. Percent distribution of non-communicable diseases, community and background characteristics of women in Ghana in 2014.**

| Characteristics | Number | Percent |
|---|---|---|
| **Prevalence of Non-Communicable Diseases** | | |
| Hypertension | 1240 | 16.16 |
| Obesity | 543 | 17.19 |
| Anemia | 1692 | 41.05 |
| **Community-level variables** | | |
| Percent of women educated | | |
| Low | 2778 | 24.24 |
| Medium and High | 5517 | 75.76 |
| Percent of poor women | | |
| Low | 2820 | 44.59 |
| Medium and High | 5475 | 55.41 |
| Percent of working women | | |
| Low | 4306 | 47.13 |
| Medium and High | 3744 | 52.27 |
| **Individual-level characteristics** | | |
| Respondent's Education | | |
| None | 2231 | 20.93 |
| Primary | 1431 | 16.72 |
| Secondary/higher | 4633 | 62.36 |
| Wealth Index | | |
| Poorest/Poorer | 3510 | 32.21 |
| Middle/Richer/Richest | 4785 | 67.79 |
| Ethnicity | | |
| Akan | 3444 | 50.73 |
| Other Tribes | 4758 | 49.27 |
| Region | | |
| Northern | 2289 | 14.16 |
| Southern | 5809 | 85.84 |
| Place of Residence | | |
| Urban | 4131 | 45.51 |
| Rural | 4164 | 54.49 |
| Age group | | |
| 18–24 | 2226 | 26.56 |
| 25–34 | 2907 | 35.49 |
| 35–49 | 3162 | 37.95 |
| Health Insurance Coverage | 5504 | 62.55 |
| Visited Health facility in last 12 months | 4543 | 54.08 |

Source: GDHS 2014; Weighted Frequencies.

On community level characteristics, 76 percent of women resided in communities with medium to high levels of education (that is with primary education and above) while 55 percent resided in communities considered poor using the wealth index of the households within that community. Similarly, 52 percent of women resided in communities with middle to high levels of women working outside the household.

Table 1 also highlights the individual level characteristics of the respondents. Sixty-two percent had a secondary or higher education, 32 percent resided in households considered poor according to the wealth index. There was an even split in ethnicity between Akan and non-Akans (51 vs 49 percent) and majority of respondents resided in the Southern part of the country (86 percent) and in rural areas (55 percent). Finally, 63 percent had access to health insurance coverage while 54 percent had visited a health facility in the last 12 months.

Table 2 presents bivariate analysis of NCDs by community level variables and individual level characteristics. The Table reports column percentages (for easy comparison across categories of NCDs) and 95 percent Confidence Intervals of the proportions. In communities with medium to high levels of poverty, 12 percent were hypertensive compared to 18 percent in communities with medium to high levels of education. Similarly, 21 percent of women in communities with medium to high levels of education were either overweight or obese compared to only 9 percent in poor communities. These relationships were statistically significant, using the Chi-square. The results of the percent of working women in the community and NCDs were not statistically significant, although in the expected direction. For example, 17 percent were hypertensive in communities with a medium to high percentage of working women (compared to 16 percent in communities with a low percentage of working women) and 22 percent of women in communities with a high percentage of working women were obese (compared to 16 percent in communities with a low percentage of working women). All the community level variables had similar risk of reporting anemia of any form (40% in communities with high levels of education (compared to 44 precent in communities with low levels of education), 42 percent in poor communities (compared to 39 percent with low levels of poverty) and 42 in communities with high percentage of working women, although as indicated, this relationship was not statistically significant). (See Table 2).

On individual women's characteristics, women in wealthier households reported higher levels of NCD risk factors than poorer households (e.g., 19 percent vs. 11 percent reported hypertension and 23 percent vs. 4 percent reported obesity). Residing in Southern Ghana was associated with higher risks of hypertension than Northern Ghana (18 percent vs. 10 percent). However, the risk was reversed for obesity-48 percent were obese or overweight in Northern Ghana compared to 20 percent in Southern Ghana. Urban areas are also associated with higher NCD risks than rural areas (e.g., 20 percent vs. 12 percent for hypertension and 23 percent vs. 10 percent for obesity). Older women were also at a higher risk of NCDs compared to younger women (e.g., 28 percent of women aged 35–49 were hypertensive (27 percent were obese/overweight) compared to only 5 percent of respondents aged 18–24 for both hypertension and obesity). Finally, women with access to health insurance (17 percent) and women who visited a health facility in the last 12 months (18 percent) reported hypertension. All the above-described relationships were statistically significant given the chi-square value.

## Multivariate results

In Table 3, we start with the multilevel logistic regression results for hypertension. The aggregate-level of education, although positive, was not a significant predictor of the prevalence of hypertension. However, the aggregate-levels of poverty and employment in the community were significantly associated with hypertension, holding all other covariates constant. The

**Table 2. Crosstabulation of non-communicable diseases by community variables and individual background characteristics of women in Ghana in 2014 (Column percentages (95% CI)).**

| | Hypertension | | Obesity | | Anemia | |
|---|---|---|---|---|---|---|
| **Community-level variables** | % (95% CI) | p-value | % (95% CI) | p-value | % (95% CI) | p-value |
| Percent of educated women | | | | | | |
| Low | 10.36 (0.087, 0.122) | | 5.17 (0.036, 0.075) | | 44.12 (0.409, 0.474) | |
| Medium and High (Versus Low Levels) | 18.02 (0.166, 0.196) | <0.001 | 20.96 (0.189, 0.232) | <0.001 | 40.05 (0.38, 0.430) | 0.050 |
| Percent of poor women | | | | | | |
| Low | 20.9 (0.192, 0.228) | | 26.63 (0.236, 0.299) | | 38.71 (0.355, 0.410) | |
| Medium and High (Versus Low Levels) | 12.32 (0.110, 0.138) | <0.001 | 9.28 (0.080, 0.107) | <0.001 | 42.94 (0.406, 0.453) | 0.050 |
| Percent of working women | | | | | | |
| Low | 15.94 (0.146, 0.174) | | 16.2 (0.144, 0.181) | | 40.8 (0.387, 0.430) | |
| Medium and High (Versus Low Levels) | 17.16 (0.149, 0.196) | 0.390 | 21.83 (0.164, 0.284) | 0.064 | 42.20 (0.376, 0.469) | 0.593 |
| **Individual-level characteristics** | | | | | | |
| Respondent's Education | | | | | | |
| None | 14.36 (0.125, 0.16) | | 9.03 (0.069, 0.118) | | 45.29 (0.411, 0.486) | |
| Primary | 17.22 (0.148, 0.199) | | 18.15 (0.146, 0.224) | | 42.28 (0.380, 0.468) | |
| Secondary/higher | 16.49 (0.151, 0.18) | 0.145 | 19.66 (0.176, 0.219) | <0.001 | 39.25 (0.369, 0.417) | 0.016 |
| Wealth Index | | | | | | |
| Poorest/Poorer | 10.49 (0.092, 0.119) | | 4.33 (0.031, 0.059) | | 45.77 (0.431, 0.484) | |
| Middle/Richer/Richest | 18.86 (0.174, 0.204) | <0.001 | 23.23 (0.298, 0.256) | <0.001 | 38.76 (0.362, 0.414) | <0.001 |
| Ethnicity | | | | | | |
| Akan | 17.02 (0.154, 0.188) | | 18.63 (0.164, 0.211) | | 38.76 (0.360, 0.416) | |
| Other Tribes | 15.42 (0.140, 0.169) | 0.135 | 15.68 (0.010) | 0.105 | 43.23 (0.475, 0.457) | 0.018 |
| Region | | | | | | |
| Northern | 10.01 (0.080, 0.124) | | 47.9 (0.025, 0.070) | | 43.03 (0.391, 0.470) | |
| Southern | 17.45 (0.1611, 0.189) | <0.001 | 19.39 (0.175, 0.215) | <0.001 | 40.73 (0.385, 0.430) | 0.317 |
| Place of Residence | | | | | | |
| Urban | 19.81 (0.183, 0.214) | | 22.97 (0.23, 0.258) | | 40.67 (0.378, 0.436) | |
| Rural | 11.79 (0.103, 0.135) | <0.001 | 9.95 (0.084, 0.1175) | <0.001 | 41.5 (0.390, 0.441) | 0.671 |
| Age group | | | | | | |
| 18–24 | 5.03 (0.040, 0.063) | | 4.69 (0.035, 0.063) | | 44.59 (0.412, 0.480) | |
| 25–34 | 12.09 (0.103, 0.142) | | 16 (0.135, 0.189) | | 39.15 (0.360, 0.425) | |
| 35–49 | 27.77 (0.256, 0.300) | <0.001 | 26.62 (0.236, 0.299) | <0.001 | 40.33 (0.374, 0.434) | 0.058 |
| Health Insurance Coverage | 17.15 (0.158, 0.185) | <0.001 | 17.6 (0.155, 0.198) | 0.563 | 41.04 (0.387, 0.4341) | |
| Visited Health facility in last 12 months | 17.81 (0.163, 0.194) | <0.001 | 18.48 (0.1637, 0.208) | 0.061 | 39.61 (0.371, 0.421) | 0.086 |

Source: GDHS 2014; Weighted Frequencies.

odds of hypertension were lower if aggregate levels of poverty are high in the community (compared to lower levels of poverty)—(Odds Ratio (OR) = 0.78, p<0.05). Following the same pattern, a high aggregate level of women's employment in the community was associated with higher odds of hypertension (Odds Ratio (OR) = 1.12, p<0.05).

On individual and household correlates of hypertension, individual women's education, ethnicity, and region of residence (South vs. the North) were not statistically significant predictors of hypertension. The odds of hypertension, however, were higher with the level of wealth of a household. The odds of hypertension among rich compared to poor households were 1.3 times (p<0.05), confirming the earlier results that poverty reduces your odds of hypertension. Older women (aged 35–49) and visiting a health facility in the last 12 months were associated

**Table 3. Odds ratios of multilevel models of community variables, covariates and non-communicable diseases among women in Ghana in 2014.**

| | Hypertension | | Obesity | | Anemia | |
|---|---|---|---|---|---|---|
| **Community-level variables** | **OR (95% CI)** | **p-value** | **OR (95% CI)** | **p-value** | **OR (95% CI)** | **p-value** |
| Percent of women educated | | | | | | |
| Low | 1 | | 1 | | 1 | |
| Medium and High | 1.03 (0.857, 1.248) | 0.726 | 1.05 (0.815, 1.36) | 0.697 | 0.977 (0.837, 1.142) | 0.772 |
| Percent of poor women | | | | | | |
| Low | 1 | | 1 | | 1 | |
| Medium and High | 0.78 (0.659, 0.924) | 0.004 | 0.59(0.443, 0.808) | <0.001 | 1.10 (0.910, 1.319) | 0.293 |
| Percent of working women | | | | | | |
| Low | 1 | | 1 | | 1 | |
| Medium and High | 1.12 (1.010, 1.237) | 0.031 | 1.17(1.022, 1.348) | 0.024 | 1.032 (0.939, 1.136) | 0.504 |
| **Individual-level characteristics** | | | | | | |
| Respondent's Education | | | | | | |
| None/primary | 1 | | 1 | | 1 | |
| Secondary/higher | 1.01 (0.897, 1.137) | 0.864 | 1.20(1.030, 1.420) | 0.021 | 0.94 (0.839, 1.046) | 0.246 |
| Wealth Index | | | | | | |
| Poorest/Poorer | 1 | | 1 | | 1 | |
| Middle/Richer/Richest | 1.30(1.038, 1.625) | 0.022 | 3.09(1.946, 4.913) | <0.001 | 0.73(0.556, 0.964) | 0.027 |
| Ethnicity | | | | | | |
| Other Tribes | 1 | | 1 | | 1 | |
| Akan | 0.91 (0.778, 1.075) | 0.280 | 0.76(0.580, 0.991) | 0.043 | 0.89 (0.741, 1.074) | 0.229 |
| Region | | | | | | |
| Northern | 1 | | 1 | | 1 | |
| Southern | 1.15+ (0.879, 1.50) | 0.310 | 1.45 (0.909, 2.314) | 0.119 | 1.36 (1.069, 1.727) | 0.012 |
| Place of Residence | | | | | | |
| Rural | 1 | | 1 | | 1 | |
| Urban | 0.85 (0.655, 1.094) | 0.202 | 0.89 (0.804, 1.591) | 0.479 | 0.78(0.630, 0.955) | 0.017 |
| Age group | | | | | | |
| 18-24/25-34 | 1 | | 1 | | 1 | |
| 35–49 | 2.83(2.499, 3.209) | <0.001 | 2.85 (2.430, 3.358) | <0.001 | 0.91 (0.820, 1.005) | 0.063 |
| Health Insurance Coverage | | | | | | |
| No | 1 | | 1 | | 1 | |
| Yes | 1.16 (0.986, 1.372) | 0.730 | 0.89 (0.681, 1.178) | 0.430 | 1.072 (0.907, 1.268) | 0.411 |
| Visited Health facility in last 12 months | | | | | | |
| No | 1 | | 1 | | 1 | |
| Yes | 1.23 (1.051, 1.437) | <0.001 | 1.16 (0.940, 1.420) | 0.168 | 0.878 (0.745, 1.04) | 0.118 |

Source: GDHS 2014.

with higher odds of hypertension (Odds Ratio (OR) = 2.83, p<0.001 and 1.23, p<0.001; respectively).

Next, we consider the multilevel results of the prevalence of obesity. Like hypertension, the aggregate level of education in the community did not significantly predict of obesity. However, the aggregate-level of poverty and women's employment in the community as observed on the model on hypertension was significantly associated with obesity. The odds of obesity were lower if aggregate levels of poverty are high in the community (Odds Ratio (OR) = 0.59, p<0.001) and odds of obesity were higher if the aggregate level of women employed outside the household was higher (Odds Ratio (OR) = 1.17, p<0.05).

On covariates, unlike hypertension, an increase in individual education had significantly higher odds of obesity (Odds Ratio (OR) = 1.20, p<0.05). The odds of obesity of women with a secondary or higher education were 1.2 times that of women who never went to school or have only a primary education. Similarly, household wealth status and women's age, as observed in the model on hypertension, had higher odds of obesity. The odds of obesity if you reside in a richer household were 3.09 times compared to poorer households. That of older women were similar; 2.85 times. Ethnicity (Akan vs other tribes) had lower odds of obesity (Odds Ratio (OR) = 0.76, p<0.05). Urban residence, having health insurance and visit to a health facility in the last 12 months were not significant predictors of obesity.

Finally, we consider the models for anemia. Community-level variables did not significantly predict the prevalence of anemia. On covariates of anemia, richer households had lower odds of anemia (Odds Ratio (OR) = 0.73, p<0.05). Similarly urban residence had lower odds (Odds Ratio (OR) = 0.78, p<0.05). Surprisingly, residence in Southern Ghana, which is socio-economically well-off than Northern Ghana increases the odds of anemia (Odds Ratio (OR) = 1.36, p<0.05).

## Discussion

NCDs are rising rapidly in Africa and undermining social and economic development. Ghana, like many middle- to low-income countries, is dealing with the triple burden of infectious diseases, malnutrition/overweight obesity, and chronic diseases. This study probed if neighborhood effects, or community variables played a role in the risks of NCDs. In both bivariate and multivariate analysis, women living in poorer communities reported lower prevalence of NCDs (hypertension and obesity), while women residing in communities with a high level of employment reported higher risks of NCDs (hypertension and obesity). These results were reinforced with the results of the wealth index of a household and the individual educational level of women. Wealthier households had higher odds of hypertension and obesity and lower odds of anemia. Women with higher education (secondary and higher level) had higher odds of obesity.

First, the results of this study are largely consistent with the literature on neighborhood effects on one hand and the nutritional and epidemiological transition as well as the globalization of unhealthy lifestyles on the other. The results confirmed that there is considerable social inequality among communities and NCD risks tend to be bundled together at the neighborhood/community level, but this is slightly inconsistent with the social gradient of health hypotheses. In developing countries like Ghana, the current NCD disease burden is greater among the middle class compared to the lower class in higher income countries. As stated above, there is a high risks of NCDs in communities with a high levels of women employed and in wealthier households, while there are lower risks in communities with high levels of poverty. It is thus urgent to develop targeted primary intervention strategies in both communities (rich and poor). As we know from the nutritional and epidemiological transitions soon, if not already, NCDs will be concentrated among both the rich and poor as rapid urbanization brings people from rural communities to the cities, like what prevails in higher income countries [47].

Second, within the context of the epidemiological and nutritional transition–which focuses on the complex change in patterns of health and disease and on the interactions between these patterns and the demographic, economic, and sociological determinants and consequences [48,49]. It is clear that developing countries like Ghana have scarcely completed the previous stages of the transition (where infectious diseases prevail) before gradually shifting to the age of degenerative and lifestyle diseases with NCDs emerging as the major causes of death.

Globalization and diffusion of unhealthy human diet and lifestyle behaviors such as eating processed Western diets and foods high in calories, sugar, salt, and total fat [37,50,51] as well as the lack of physical exercise and immoderate alcohol consumption has occurred in parallel with the increasing prevalence of obesity and NCDs in low-middle-income countries [51,52]. There is thus a need to design and fund effective awareness and early intervention programs to mitigate the human resource and economic burden of NCDs while tackling the continuing scourge of infectious diseases and malnutrition. This will accelerate the attainment of the 2030 Agenda for Sustainable Development (SGDs) and a demographic dividend. This study by evaluating the significance of key community variables and individual determinants of NCD risks in low-middle-income countries, has underscored the importance of country-tailored assessments that are needed for awareness, detection, and management of NCDs.

Finally, our study is not without limitations. Apart from its cross-sectional design, the differential selection of individuals into communities and other indirect pathways of neighborhood/community factors or simultaneity biases prevent us from drawing causal inferences [53–55]. Similarly, our study is unable to clarify if neighborhood/community differences are due to characteristics of the areas or differences in the types of people living in different areas. Thus, there is the need for further research to distinguish between context and composition when examining neighborhood effects on health. Nonetheless, neighborhood-level mechanisms have proven effective in other outcomes and can be usefully applied to NCDs with important policy implications for health promotion and reduction of health disparities in low-middle-income countries.

## Author Contributions

**Conceptualization:** Winfred A. Avogo.

**Data curation:** Winfred A. Avogo.

**Formal analysis:** Winfred A. Avogo.

**Methodology:** Winfred A. Avogo.

**Writing – original draft:** Winfred A. Avogo.

**Writing – review & editing:** Winfred A. Avogo.

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
