## [Decision Letter · Decision Letter 0]

13 Apr 2022

PGPH-D-21-01199

Community Characteristics and the Risk of Noncommunicable Diseases in Ghana

Dear Dr. Avogo,

Thank you for submitting your manuscript to PLOS Global Public Health. After careful consideration, we feel that it has merit but does not fully meet PLOS Global Public Health’s publication criteria as it currently stands. Therefore, we invite you to submit a revised version of the manuscript that addresses the points raised during the review process.

We look forward to receiving your revised manuscript.

Kind regards,

Jasper Tromp

Academic Editor

Journal Requirements:

1. Please update the completed 'Competing Interests' statement, including any COIs declared by your co-authors. If you have no competing interests to declare, please state "The authors have declared that no competing interests exist". 

2. Please amend your detailed Financial Disclosure statement. If you did not receive any funding for this study, please simply state: “The authors received no specific funding for this work.”

3. Please include the direct link of othe site where your data can be downloaded. Please update the Data Availability Statement.

Additional Editor Comments (if provided):

I recommend to

1. Significantly shorten the discussion, particularly the literature discussion section should be combined with the introduction

2. Provide more information on your multivariable analysis. What variables did you select as confounders? Why?

3. Include 95% confidence bounds for all your reported odds ratios.

4. Significantly revise the discussion: for example, the second paragraph is simply repeating your results. This is unnecessary. Try to highlight your key findings.

Reviewers' comments:

Reviewer's Responses to Questions

**Comments to the Author**

1. Does this manuscript meet PLOS Global Public Health’s publication criteria? Is the manuscript technically sound, and do the data support the conclusions? The manuscript must describe methodologically and ethically rigorous research with conclusions that are appropriately drawn based on the data presented.

Reviewer #1: No

Reviewer #2: Yes

2. Has the statistical analysis been performed appropriately and rigorously?

Reviewer #1: No

Reviewer #2: No

3. Have the authors made all data underlying the findings in their manuscript fully available (please refer to the Data Availability Statement at the start of the manuscript PDF file)?

Reviewer #1: No

Reviewer #2: Yes

4. Is the manuscript presented in an intelligible fashion and written in standard English?

Reviewer #1: No

Reviewer #2: Yes

5. Review Comments to the Author

Reviewer #1: 1. The introduction and the literature review section should be concise. There is unnecessary details which will distract the audience.

2. The manuscript must be edited by a native English editor and proofread thoroughly.

3. How cluster effect was adjusted and multicollinearity was assessed.

Reviewer #2: In this study, Winfred A. Avogo, PhD single-handedly investigates an important topic on community characteristics and the risk of non-communicable diseases in Ghana. I would have expected this work to be collaborative to constitute a few other contributors given the scope of the subject as well as the type of data that has been utilised. Nonetheless, the paper is well written, and I think there are some issues that the author should address:

Introduction

1. First, I think the Introduction is disproportionately longer considering the overall length of the article. Concerning this, is the section “Literature Review and Theory” a necessary addition to the Introduction? Are there opportunities to succinctly mention all important concepts/theories in the Introduction?

2. Also, I do not find sufficient justification for the focus on women or the reasons for excluding men from the analysis.

Methods

1. On page 12, you state that you used two variables to capture aggregate levels of education and employment in the community. But, in essence, there are three variables mentioned in that paragraph. Please clarify.

2. On page 13, I got confused by this statement “We use individual educational attainment (1. No education, 2. Primary education, Secondary or higher education) as a covariate to examine the impact of community variables on the dependent outcomes, controlling for individual education”. This might need some revision for clarity.

Results

1. Table 1. Is possible to make the heading of the table more informative?

2. Is it possible to label which variables were “individual-level characteristics”?

3. Most point estimates, e.g. the prevalence of NCDs or proportions in general, consistently lack the confidence intervals. Was it impossible to do the calculations?

4. There was inconsistency in the number of decimal points?? Compare 17 percent and 17.19 percent on page 14.

5. Is 16.16% nearly 16% or 17%?

6. Table 2: What do the values in each cell represent? Put in another way, what are you comparing to what? (Refer to paragraph 3, line 1 page 15). Why are some cells empty for community-level variables. Maybe, the title is not informative as well to help the reader interpret the table correctly. Again, there is a need to demarcate where individual-level variables start from in the table.

7. So, how did you determine if the relationships were statistically significant in the absence of p-values or confidence intervals? Refer to the last line, paragraph 3, page 15.

8. Check the title for Table 3 for a possible typo “coviates”.

9. I think it would be ideal to choose a single cut-off p-value for consistency, for example, p<0.05. Comparisons become difficult when you have used three cut-off p-values. This might sort out the issue of “marginal significance” where the author attempts to ensure some factors appear significant, yet they are not. I would avoid this; it is either the variables are statistically significant or not; there is no borderline.

10. Minor observations. There are instances where the author uses the present continuous tense for events that have already occurred.

6. PLOS authors have the option to publish the peer review history of their article (what does this mean?). If published, this will include your full peer review and any attached files.

**Do you want your identity to be public for this peer review?** For information about this choice, including consent withdrawal, please see our Privacy Policy.

Reviewer #1: **Yes: **Rajat Das Gupta

Reviewer #2: No

---

## [Decision Letter · Decision Letter 1]

28 Jul 2022

PGPH-D-21-01199R1

Community Characteristics and the Risk of Noncommunicable Diseases in Ghana

Dear Dr. Avogo,

Thank you for submitting your manuscript to PLOS Global Public Health. After careful consideration, we feel that it has merit but does not fully meet PLOS Global Public Health’s publication criteria as it currently stands. Therefore, we invite you to submit a revised version of the manuscript that addresses the points raised during the review process.

We look forward to receiving your revised manuscript.

Kind regards,

Rajat Das Gupta, M.D.

Academic Editor

Journal Requirements:

Additional Editor Comments (if provided):

1. The manuscript needs to be proofread and edited by a native English speaker.

2. This sections are not still clear in the methodology section:

a. The authors did not mention how the variables were adjusted in the final multivariable model?

b. The authors excluded pregnant women from BMI, why they did not exclude who delivered within preceding 2 months of the survey? This is to exclude Pregnancy related weight gain from true overweight/obesity.

c. In the hypertension definition, those who were taking anti-hypertensive medications but had BP within limit were they classified as hypertensive?

Reviewers' comments:

Reviewer's Responses to Questions

**Comments to the Author**

1. If the authors have adequately addressed your comments raised in a previous round of review and you feel that this manuscript is now acceptable for publication, you may indicate that here to bypass the “Comments to the Author” section, enter your conflict of interest statement in the “Confidential to Editor” section, and submit your "Accept" recommendation.

Reviewer #1: All comments have been addressed

Reviewer #2: All comments have been addressed

Reviewer #3: (No Response)

2. Does this manuscript meet PLOS Global Public Health’s publication criteria? Is the manuscript technically sound, and do the data support the conclusions? The manuscript must describe methodologically and ethically rigorous research with conclusions that are appropriately drawn based on the data presented.

Reviewer #1: Yes

Reviewer #2: Yes

Reviewer #3: Partly

3. Has the statistical analysis been performed appropriately and rigorously?

Reviewer #1: Yes

Reviewer #2: Yes

Reviewer #3: No

4. Have the authors made all data underlying the findings in their manuscript fully available (please refer to the Data Availability Statement at the start of the manuscript PDF file)?

Reviewer #1: Yes

Reviewer #2: Yes

Reviewer #3: Yes

5. Is the manuscript presented in an intelligible fashion and written in standard English?

Reviewer #1: Yes

Reviewer #2: (No Response)

Reviewer #3: Yes

6. Review Comments to the Author

Reviewer #1: The authors have addressed all the comments.

Reviewer #2: Abstract

In the abstract, please delete the term “recent” because I don’t think data collected in 2014 is recent. However, it is understood that there is a paucity of data on this subject and that the data/study is still relevant and important for publication.

The introduction and methods sections are well revised and clear

However, there are a few edits which I think might be necessary for the Results and Discussion sections:

1. In Table 1, it is necessary to indicate where and when the study was conducted, for example, you can add the phrase “…in Ghana in 2014”. Table and Figure headings usually require such information.

2. In table 2, there is a mismatch with the information provided in the text, page 13—paragraph 2. The text states that the values in the cells are percentages (proportions) while the titles (the third row, Table 2) indicate OR (95% CI). Please clarify

3. Still in Table 2, whether the point estimates are proportions (percentages) or odds ratios (but I guess they are proportions based on information provided in the text), you may want to indicate the reference categories, in brackets. For example, Percent of women with medium or high levels of education (versus low levels), etc.

4. Still, in Table 2, you might want to change all p-values indicated ad “0.000” to be “<0.001” if 3 decimal places are required by journal guidelines. This applies to all to table 3 as well.

5. Page 13, paragraph 2. “…using the Chi-square test of independence?” Mention this in the Methods section as well.

6. Discussion paragraph 1, “…reported a lower incidence of NCDs…” I don’t this study measured incidence, but rather prevalence and risk factors. So, please replace the term “incidence” with an appropriate term.

Reviewer #3: Document uploaded

7. PLOS authors have the option to publish the peer review history of their article (what does this mean?). If published, this will include your full peer review and any attached files.

**Do you want your identity to be public for this peer review?** For information about this choice, including consent withdrawal, please see our Privacy Policy.

Reviewer #1: No

Reviewer #2: **Yes: **Dr Jonathan A Abuga (MPH, PhD)

Reviewer #3: No

---

## [Decision Letter · Decision Letter 2]

17 Nov 2022

PGPH-D-21-01199R2

Community Characteristics and the Risk of Non-communicable Diseases in Ghana

Dear Dr. Winfred Avogo,

Thank you for submitting your manuscript to PLOS Global Public Health. After careful consideration, we feel that it has merit but does not fully meet PLOS Global Public Health’s publication criteria as it currently stands. Therefore, we invite you to submit a revised version of the manuscript that addresses the points raised during the review process.

We look forward to receiving your revised manuscript.

Kind regards,

Rajat Das Gupta, M.D.

Academic Editor

Journal Requirements:

2. We do not publish any copyright or trademark symbols that usually accompany proprietary names, eg  ©, ®, ™  (e.g. next to drug or reagent names). Please remove all instances of trademark/copyright symbols throughout the text, including ® on page 9.

Additional Editor Comments (if provided):

Reviewers' comments:

Reviewer's Responses to Questions

**Comments to the Author**

1. If the authors have adequately addressed your comments raised in a previous round of review and you feel that this manuscript is now acceptable for publication, you may indicate that here to bypass the “Comments to the Author” section, enter your conflict of interest statement in the “Confidential to Editor” section, and submit your "Accept" recommendation.

Reviewer #2: All comments have been addressed

2. Does this manuscript meet PLOS Global Public Health’s publication criteria? Is the manuscript technically sound, and do the data support the conclusions? The manuscript must describe methodologically and ethically rigorous research with conclusions that are appropriately drawn based on the data presented.

Reviewer #2: Yes

3. Has the statistical analysis been performed appropriately and rigorously?

Reviewer #2: Yes

4. Have the authors made all data underlying the findings in their manuscript fully available (please refer to the Data Availability Statement at the start of the manuscript PDF file)?

Reviewer #2: Yes

5. Is the manuscript presented in an intelligible fashion and written in standard English?

Reviewer #2: Yes

6. Review Comments to the Author

Reviewer #2: Table 2: For community level variables, please indicate the proportions/percentages for all categories, for instance, for percent of educated women, show the proportions for both “medium and high” and “low levels.

7. PLOS authors have the option to publish the peer review history of their article (what does this mean?). If published, this will include your full peer review and any attached files.

**Do you want your identity to be public for this peer review?** For information about this choice, including consent withdrawal, please see our Privacy Policy.

Reviewer #2: No

---

## [Editor Report · Decision Letter 3]

13 Dec 2022

Community Characteristics and the Risk of Non-communicable Diseases in Ghana

PGPH-D-21-01199R3

Dear Dr. Avogo,

We are pleased to inform you that your manuscript 'Community Characteristics and the Risk of Non-communicable Diseases in Ghana' has been provisionally accepted for publication in PLOS Global Public Health.

Best regards,

Rajat Das Gupta, M.D.

Academic Editor